# A Personal Prospective on Testosterone Therapy in Women—What We Know in 2022

**DOI:** 10.3390/jpm12081194

**Published:** 2022-07-22

**Authors:** Gary S. Donovitz

**Affiliations:** 1Morehouse School of Medicine, Department of Obstetrics and Gynecology, Atlanta, GA 30310, USA; gary@donovitz.com; 2BioTE Medical, LLC, 1875 West Walnut Hill Lane, Suite 100, Irving, TX 75038, USA

**Keywords:** hormone replacement therapy, testosterone deficiency, compounded bioidentical hormone therapy

## Abstract

Hormone replacement therapy continues to be a controversial topic in medicine, with certain narratives regarding safety concerns that are not scientifically established in peer-reviewed literature. These negative narratives, specifically undermining the use of testosterone in women, have caused women to remain without any Food and Drug Administration (FDA)-approved testosterone therapies, while more than 30 FDA-approved testosterone therapies are available for men in the United States. This has resulted in millions of women suffering in silence with very common symptoms of perimenopause and menopause that could easily be addressed with the use of testosterone. There is growing evidence to support the use of physiologic doses of testosterone for sexual function, osteoporosis prevention, brain protection, and breast protection. The safety of testosterone use in women has been evaluated for the past 80 years. A recent publication on the complications of subcutaneous hormone-pellet therapy, looking at a large cohort of patients over 7 years, demonstrated long-term safety. In addition, there have been two large long-term peer-reviewed studies showing a significant reduction in the incidence of invasive breast cancer in women on testosterone therapy. Perhaps it is time for the FDA to consider approving products that would benefit testosterone-deficient women.

## 1. Commentary

Hormone supplementation and/or optimization continues to be an extremely controversial area of medicine, with certain clinicians promoting inaccurate narratives regarding safety concerns that are not truly validated in the scientific peer-reviewed literature.

## 2. Women’s Health Initiative (WHI) and Its Impact on Hormone Replacement Therapy (HRT)

The findings of the conjugated equine estrogens (CEE) and medroxyprogesterone acetate (MPA) arms of the WHI were published in 2002 and dramatically changed the prescribing practices of physicians in the United States [1]. The trial demonstrated adverse cardiovascular disease events and an increased risk of breast cancer, deep vein thrombosis, and Alzheimer’s disease in female patients in the combined estrogen plus progestin (E + P) arm, but not in the CEE-only arm. The number of women for whom estrogen and progestin are prescribed had been steadily increasing, from 58 million in 1995 to 90 million in 1999 [1]. After the publication in 2002, prescriptions for HRT plummeted by more than 60%.

Since the WHI studies almost twenty years ago, much confusion has occurred regarding estrogen supplementation, and this has led to many negative and inaccurate perceptions around the use of testosterone in women.

## 3. History of Testosterone in Women

Testosterone has been used in women for over 80 years to treat perimenopause and menopause symptoms [2]. In England and Australia, testosterone has been licensed for use in women for more than 60 years.

The role of androgens in female health and well-being is a topic of growing interest, with hundreds of thousands of women in the United States having their testosterone optimized and reporting improvements in their quality of life and general health. Notwithstanding this, controversy exists concerning the existence of androgen-deficiency states and their clinical diagnosis and management. Historically, androgens have been associated with masculinity or male sexual function, which has undoubtedly contributed to a lack of recognition of androgen effects in women. In fact, androgens are necessary not only for the development of reproductive function and hormonal homeostasis in women but also represent the immediate precursors for the in vivo synthesis of estrogens [3].

## 4. Androgen Deficiency in Women

Following a bilateral salpingo-oophorectomy, a precipitous decline in estrogen occurs in women, and for years, a multitude of estrogen replacement therapies have been FDA-approved and utilized by practitioners. The simultaneous decline in serum testosterone levels has received far less attention. Serum androgen levels decline steeply in the early reproductive years. A study by Glaser et al. effectively documents that testosterone can improve most common post-menopausal symptoms [4]. Pre- and post-menopausal patients may experience symptoms of androgen deficiency, including hot flashes, night sweats, decreased libido, irritability, anxiety, depression (also known as dysphoric moods), fatigue, a decreased feeling of well-being, poor memory, poor focus and concentration, insomnia, joint pains, vaginal dryness, urinary complaints, and incontinence, all of which are becoming increasingly recognized by practitioners. A double-blind randomized trial demonstrated that androgens affect sexual desire, bone density, muscle mass and strength, adipose tissue distribution, mood, energy, and psychological well-being [5]. It therefore stands to reason that an imbalance in androgen biosynthesis or metabolism in women may have undesirable effects on any or all of these domains.

According to Panay and Fenton [6], young women’s ovaries produce approximately three to four times more testosterone than estrogen daily. In 2002, Dimitrakakis et al. [7] stated that testosterone is the most abundant biologically active gonadal hormone throughout the female lifespan. However, unfortunately, due to a plethora of misconceptions, women remain without any FDA-approved testosterone therapies, while more than 30 approved testosterone therapies are available for men. This has resulted in millions of women suffering in silence with very common symptoms that could easily be addressed with the use of testosterone.

## 5. Testosterone Optimization in Women

There are data [8] to support the use of testosterone in patients suffering from sexual dysfunction and, more specifically, from what is termed hypoactive sexual desire disorder (HSDD). In Islam’s study, there were 36 RCTs encompassing 8480 patients, showing that testosterone use benefited women with HSDD. In addition, this meta-analysis demonstrated that non-oral testosterone was better at maintaining neutral lipid profiles than was oral testosterone [9]. HSDD is a sexual disorder characterized by distress related to a loss of or decline in sexual interest. It is estimated to affect approximately one in ten women. Menopausal status has a significant impact on the prevalence of HSDD, with several studies showing that the prevalence of HSDD is greatest in younger, surgically menopausal women (16–26%) compared with naturally pre-menopausal women (7–14%). This most likely relates to the significant decrease in testosterone in surgically induced menopausal women [10]. Dr. Davis and her consensus panel agreed that HSDD is a clinical diagnosis and that, therefore, serum testosterone levels should not be used to make a diagnosis [11]. The panel, on the one hand, recommended that treatment with testosterone should attempt to mirror pre-menopausal serum levels to achieve the desired clinical response, but then recommended using testosterone products in female patients that are FDA-approved products for males.

## 6. Route of Administration—Does It Matter?

There are many ways to prescribe testosterone, including transdermal, oral, intramuscular, and subcutaneous pellet implant application. I have had many years of experience with sub-cutaneous administration of testosterone and estrogen with the use of pellets, and there are peer-reviewed publications to support this type of administration [4]. Compounded pellets were first described in 1950 as a treatment to address menopausal symptoms [2]. The subcutaneous administration of testosterone has been used worldwide for decades. Practitioners have found this route of administration to provide safe and effective therapy without the fluctuations in blood levels often seen after transdermal [12] or intramuscular administration.

The injectable and transdermal male preparations, however, have been shown to cause elevated spikes in serum testosterone levels and an increased risk of side effects and secondary reactions [13]. The pooling of the risks from three RCTs showed 37% greater total androgenic adverse events compared with the placebo group (RR: 1.37; 95% C.I.: 1.12, 1.69; *p* = 0.002). Unfortunately, there have been no randomized head-to-head trials comparing transdermal delivery with sub-cutaneous pellets. There was, however, a 7-year retrospective study in which 1,200,000 subcutaneous implant procedures were performed in 400,000 patients. The overall continuation after two insertions was 93% (C.I. 90–95). The overall complication rate was <1% [14].

## 7. Testosterone for Women—Is It Valuable for More Than HSDD?

In addition, androgens act on multiple tissue and receptor sites. One of the major organs that testosterone has a beneficial effect on is the central nervous system. Some of the biochemical reactions in the brain that have been associated with Alzheimer’s disease include an increase in beta amyloid, a decreased brain cell glucose metabolism, and a reduction in blood flow to the brain. Studies have shown that in women, both testosterone and estradiol can counter many of these and thus reduce beta amyloid deposition, improve the brain’s ability to metabolize glucose, and improve blood flow [15,16,17].

Most peer-reviewed publications about osteoporosis and testosterone, and the most cited related papers, have discussed osteoporosis in men. Approximately 10 million men and women in the United States have osteoporosis, a metabolic bone disease characterized by low bone density and the deterioration of bone architecture, which increases the risk of fractures [18]. A total of 80 percent of these patients are female. One in seven women will develop osteoporosis after the age of 50. In those who have a significant decrease in their bone mineral density (BMD), more than 50% will sustain a fracture in their lifetime. The major cause of osteoporosis in women has erroneously been thought to be estrogen deficiency due to menopause, while in men, age-related testosterone deficiency [19]. Osteoporosis medications include bisphosphonates, receptor activator of nuclear factor kappa-B ligand inhibitors, estrogen agonists/antagonists, parathyroid hormone analogues, and monoclonal antibodies [20,21]. This list, while extensive, is missing a hormone that has been used for decades to treat osteoporosis: testosterone. Almost all studies demonstrate that androgen upregulates the expression of androgen receptors in osteoblasts [22]. However, the evidence generally suggests that androgens stimulate the differentiation of osteoblasts. Mature osteoblasts have been shown to increase bone formation and increase BMD in both women and men. The 2012 Endocrine Society osteoporosis guidelines also recommend the use of testosterone therapy in men with symptomatic low testosterone who are at high risk of fracture but fail to mention women.

Testosterone receptors (TR) have also been identified in other peripheral sites, such as breast, skeletal muscle, adipose, and genital tissues. However, many physicians continue to promote narratives that we should [23] continue to hold back on treating these patients because of the unknown effects of testosterone, and that if they are treated, they should only be treated with FDA-approved preparations used in men.

The argument behind these statements is based on (1) an erroneous belief that bioidentical hormones are not regulated, (2) a lack of awareness of the published data on the safety and efficacy of testosterone, (3) the preference to claim that the administration of testosterone should only be transdermal, as sub-cutaneous administration via pellets is dangerous, and (4) the FDA’s view that since there have been risks with estrogen and progesterone, there will be risks with testosterone.

## 8. Bioidentical vs. Synthetic Hormones

To fully understand this argument, you must first understand the difference between a bioidentical hormone and a synthetic hormone. Bioidentical hormones are man-made hormones derived from plants, often soy and yams, that are chemically identical to those the human body produces. As such, they bind to the estrogen and testosterone receptors in the same way as women’s endogenous hormones have throughout their lives [24]. The U.S. FDA has approved some forms of manufactured bioidentical hormones, including bioidentical estrogens and progesterone. However, the FDA has not approved any custom-compounded bioidentical hormones. Synthetic hormones are more likely to be animal-based and different in molecular structure from women’s endogenous hormones.

## 9. Compounded Bioidentical Hormone Regulation

The statement that bioidentical hormones are not regulated is inaccurate. There currently exist long-standing regulations at the Federal and State levels. The Drug Quality and Security Act in 2014 gave the FDA additional powers to regulate compounding pharmacies, including establishing the requirement of compliance with Current Good Manufacturing Practice (cGMP) [25], as is required by the pharmaceutical industry. This allowed for improved testing for manufacturing processes, assuring purity, potency, and quality testing, in addition to improved sterility protocols.

## 10. Testosterone Therapy Dosing—How Much Testosterone Should You Give Women?

Another significant area of controversy is the question of what a normal blood level of testosterone is, and what should be used to guide clinicians in its administration. For the thousands of practitioners who have utilized testosterone therapy for testosterone deficiency (TD) in females, the standard of care is to evaluate and treat TD as a clinical syndrome that is not tied to specific laboratory values.

The best approach to dosing is to adopt an individualized dosing algorithm. There cannot be a one-size-fits-all or one-dose-fits-all approach. The goal should be to individualize the dosing to resolve the clinical symptoms, while keeping serum levels low enough to minimize androgenic side effects. In the study “Low complication rates of testosterone and estradiol implants for androgen and estrogen replacement therapy in over 1 million procedures” [14], a proprietary dosing algorithm was used to achieve testosterone levels between 150 and 250 ng/dL. This individualized approach was successful in resolving symptoms with minimal side effects. Additional long-term dosing studies are being performed at this time.

Both the Global Consensus of Testosterone use in Women [23] and the International Society for the Study of Women’s Sexual Health (ISSWSH) practice guidelines make the point that the testing of testosterone levels in women is not diagnostic and should be used only to establish a baseline. They also allow for the testing of levels to monitor therapy. The North American Menopause Society (NAMS) concurs, stating that hormone testing in general has very limited use in menopause and is usually employed to evaluate whether there is poor absorption if there is no symptom relief [26]. These opinions are not new. The Princeton Consensus Group, in 2002 [27], published their opinion that there are no age-specific normal testosterone values established for women. This is consistent with the International Consensus’ paper on Testosterone Deficiency and Treatment in Women, which states that TD is a clinical syndrome with its foundation firmly rooted in the multiple symptoms that comprise this widespread syndrome [28].

## 11. Side Effects and Secondary Reactions

The main reason cited for not using pellet therapy concerns the adverse events that can occur with testosterone. These include acne and hair growth, as well as the inability to immediately reverse the therapy. While I certainly acknowledge that these adverse events can occur, they are easily treatable. For the majority of patients, the benefits far exceed these secondary reactions, and this mode of administration has demonstrated benefits in the areas of bone health and breast cancer protection that have not been shown in studies of other routes.

Another common argument against bioidentical testosterone and estrogen use is the concern about breast and uterine cancer. There is no evidence that testosterone administration promotes endometrial cancer. It is a well-known fact, however, that unopposed estrogen, regardless of how it is administered, can promote endometrial hyperplasia and/or adenocarcinoma [29]. The failure to administer oral progesterone in women with a uterus on estrogen replacement therapy remains a clinical mistake that leads to hyperplasia and, potentially, adenocarcinoma of the endometrium.

In the United States, 240,000 women develop breast cancer annually, and 40,000 will die from the disease [30]. The lifetime risk of developing breast cancer is one in eight. Testosterone has been inaccurately implicated in the development of pre-malignant and malignant breast cells. The evidence to the contrary shows that testosterone may be protective against breast cancer. It has been known for over 70 years that it is anti-proliferative in the breast and inhibits the stimulatory actions of estrogens. Studies have shown that testosterone has been successfully used to treat breast cancer [31]. A recent study, The Dayton Study, noted that testosterone was associated with a 39% lower incidence of breast cancer than predicted by Surveillance Epidemiologic End Result (SEER) data. The conclusions are based on the treatment of over 1200 women with testosterone, with follow-up for 10 years [32]. Another more recent long-term cohort study, entitled the Testosterone Therapy and Breast Cancer Incidence Study, followed over 2300 pre- and post-menopausal women receiving testosterone with and without estrogen. The results demonstrated that the incidence of breast cancer was 40% lower than predicted by the SEER data [33].

## 12. Recent Publications—Perceptions May Not Align with Realities of Testosterone Hormone Use in Women

Now, I would like to address multiple recent publications in the literature that have promoted these inaccurate perceptions. The first is a recent clinical commentary published in *Obstetrics & Gynecology* by Dunsmoor-Su et al. [34]. There were several inaccuracies presented in this paper.

This commentary was written by physicians who are not experts in the field of testosterone therapy in women. Unfortunately, when we see outliers who have had complications, it is easy to extrapolate and reach conclusions about the therapy that are not supported by the larger group of patients whose health and quality of life have benefited from testosterone therapy. Their paper was a dialogue written by “cherry picking” references to promote the narrative that testosterone supplementation, especially when administered with sub-cutaneous pellets, should be abandoned.

Dunsmoor-Su et al. claim that their group of obstetrician gynecologists “diagnose several endometrial cancers a year from the use of estrogen and testosterone pellets that result in super-physiologic hormone levels without adequate progesterone use”. I find this perplexing, as it is the unopposed estrogen, rather than the testosterone, that can lead to the increased incidence of adenocarcinoma of the endometrium [35]. Although obstetricians and gynecologists have been taught how and when to utilize progesterone, many have failed to do so, even with FDA-approved HRT. There is no reason to entangle this problem, which has plagued us all for years, with the separate web we must traverse to utilize testosterone safely in women.

The only reference cited in their paper regarding negative outcomes with pellet therapy is a recently published retrospective chart review, in which 539 patients were on pellet hormone therapy (PHT) and 155 were on FDA-approved hormone therapy (FHT) out of a pool of 11,862 patients [36]. It is meaningless to compare cohort groups in terms of side effects because only 4.5% of the patients in the FHT group received testosterone, versus 99% of the women in the PHT group.

They also reference the National Academies of Sciences, Engineering, and Medicine’s review of the use of compounded bioidentical hormone therapy (cBHT). The NASEM consensus report in 2020 [37] concluded that “there is insufficient evidence to support the clinical utility of cBHT.” As a presenter to the committee, I find serious flaws in the NASEM report. Out of more than 3500 studies that were given to the committee to be reviewed, only a very small fraction, 13 studies, were considered in their final recommendations.

## 13. Society Consensus and Position Statements—Misinformation Has Again Led to Confusion in Women’s Health

Finally, multiple consensus statements from a variety of societies have consistently stated that bioidentical hormones should not be used, and that only FDA-approved preparations should be used. However, these opinions are not substantiated or validated with any scientifically published data.

The North American Menopause Society stated in their position statement that compounded hormone therapies are prepared by a compounding pharmacist using a provider’s prescription, and may combine multiple hormones (estradiol, estrone, estriol, dehydroepiandrosterone (DHEA), testosterone, or progesterone), use untested, unapproved combinations or formulations, or be administered via nonstandard (untested) routes such as subdermal implants, pellets, or creams [38]. Whereas some compounded hormones have not been tested, others have had extensive testing, and have published data and FDA regulation of their outsource manufacturing. While many of the compounded creams do not have published bone and breast data, subcutaneous pellet therapy studies have been published in peer-reviewed scientific journals.

The treatment for longevity and long-term quality of life is no longer an indication for menopausal hormone therapy, although peer-reviewed studies have shown that testosterone is superior for general health, somatic complaints, and many psychological symptoms [39]. The society consensus and position statements have limited HRT to the short-term treatment of menopausal symptoms. The reduction in estrogen replacement therapy that occurred after WHI led to tens of thousands of deaths in women according to one study [40].

In conclusion, there is growing evidence in support of using individualized doses of testosterone for sexual function, osteoporosis prevention, and breast protection. The safety has been evaluated for the past 80 years. Our recent publication on the complications of subcutaneous hormone-pellet therapy, looking at a large cohort of patients over 7 years, demonstrated long-term safety. Additional double-blind randomized prospective clinical studies on the long-term benefits, cardiovascular and metabolic outcomes, and side effects of testosterone use in women are needed. The FDA and pharmaceutical companies may want to consider testosterone replacement that would benefit testosterone-deficient women.

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
