# Peer review of "A Personal Prospective on Testosterone Therapy in Women—What We Know in 2022"

_jpm, 2022, doi:10.3390/jpm12081194_

Round 1
Reviewer 1 Report
This is of course a highly important topic but I feel that the style in at times confrontational and there is much reliance on personal opinion and experience rather that public trials. I do not feel that the suggestion of male bias is justified and counter arguments can be made for example
1. there are no licensed oestrogens for men, so does gender matter?
2. Menopause is a normal event for women and there are more HRT preparations licensed and higher prescribing rates. In the UK as from 2021, HRT for women is free of charge and yet men pay for TRT even if secondary to cancer therapy. T, if prescribed off label is still free of charge. This is hardly gender bias.
3. There is no licensed T formulation in the UK for women. A low dose patch was licensed briefly about 10 years ago but the indications were so narrow that it was not financially viable.
4. The lack of Pharma investment in T products for women is related to high cost and low return as male products, soon to be generic would be used at 10% of the dose and 10% cost. This is a commercial reason not gender bias.
5. There is no testosterone utopia for men as they have to fight there way past a resistance within the endocrine world not based on evidence and men suffer as well as women. The safety issues do not seem to go away and generally women are much more risk averse than men.
Author Response
I appreciate your time in reviewing my paper. I improved the style of manuscript by citing more scientific publications.
I agree with your comments that the gender bias portion of my paper may be confusing. The reasons for no testosterone products for women in the United States is multi-factorial. There is FDA hesitancy to approve such products but also as you mentioned there are commercial reasons.
I have taken your suggestion and revised the manuscript.
Again, thank you for your help in making my paper better.
Gary Donovitz MD
Reviewer 2 Report
This commentary focuses on a very important subject, which is the undertreatment of androgen deficiency in women. I agree with the author that testosterone replacement could have a beneficial effect for women with hypoactive sexual disorder as well as for women with persistent menopausal symptoms despite adequate estrogen replacement.
However, I respectfully disagree with the author's conclusions that:
- physicians should target pharmacologic and not physiologic use of testosterone
- testosterone pellets are the preferred route of administration
- testosterone use in women is a therapeutic practice with proven safety
Throughout the manuscript, several assumptions of the author seem to be based on his large personal clinical experience and not the peer-reviewed literature, which is often mentioned but not displayed.
1. The author cites the Cochrane systematic review of 2005. Since then, there has been a metanalysis of randomized controlled studies (Islam RM, et al. Lancet Diabetes Endocrinol. 2019. PMID: 31353194), which clearly merits a mention in such an article. Despite showing an acceptable safety profile, this study did show unfavorable lipid changes (HDL reduction), a finding that is not at all discussed in the current article
2. As the authors knows very well, observational and in particular retrospective studies are prone to several bias. The only randomized controlled trial with use of testosterone subcutaneous pellets is the one of Davis et al, Maturitas. 1995, PMID: 7616872 and it was in combination with estrogens. The study showed favorable effect on sexual function and bone without unfavorable effect on lipids, a finding possibly due to concomitant estrogen administration.
3. There are no long data safety outcomes (cardiovascular and metabolic outcomes) in a controlled setting outside studies of very short follow-up (< 1-2 years). This very important issue is not at all raised throughout the manuscript. We need prospective studies with longer follow-up before recommending the use of testosterone in women in broader indications than reduced libido.
4. Lines 144-148. I do not agree with the suggestion of the author that maybe supraphysiologic pharmacological doses of testosterone are needed for some women. I think that measuring testosterone levels while treated to avoid supratherapeutic levels is crucial. There is a clear sexual dimorphism regarding androgens and metabolic syndrome. High testosterone is clearly associated with metabolic syndrome and type 2 diabetes in women and polycystic ovary syndrome is a clear undisputed example. In my opinion and unless evidence from high-quality studies show otherwise, we should avoid high testosterone levels in menopausal women. This is also in agreement with Endocrine Society guidelines (Wierman, J Clin Endocrinol Metab 2014, PMID: 25279570).
5. Lines 89-90. The author repeatedly states that transdermal testosterone is associated with highly variable plasmatic T levels and high spikes as compared to testosterone pellets. I could not find any data in favor of this statement. The reference 11 cited by the author does not support this assumption. To my knowledge, there are no head-to-head comparison of plasma testosterone levels in women receiving transdermal testosterone vs subcutaneous pellets. If this is not true, the author should discuss these data in detail in order to support his claim.
6. Reference 12 is inaccessible. The link notes that the page is not found.
7. Reference 14 is not a scientific paper. It should be replaced by a peer-reviewed publication.
8. Reference 26 is misleading. It refers to a paper that compared estrogen preparations and not testosterone ones.
9. Reference 29-30 are also misleading. I agree with the author that there are no alarming data that testosterone increases breast cancer risk. Nevertheless, both of these articles combined testosterone with aromatase inhibitors that are known to reduce estrogen levels and thus decrease breast cancer risk.
Author Response
I appreciate the perspective and time that you have taken to review my manuscript. Your feedback and suggestions helped to improve my paper.
I believe as you do that checking testosterone levels is important. Although testosterone deficiency in women is a clinical syndrome, I agree checking levels and avoiding adverse side effects and metabolic complications are critical. I have incorporated your suggestions into the body of the text.
Dr. Islam's report was a great study of meta analysis of multiple RCT's and I have incorporated it into my paper .
In addition, at your suggestion I removed the pharmacologic rational for testosterone dosing and replaced it with individualized dosing.
I could not find where I stated that testosterone therapy has proven safety. I did cite a 7 year study evaluating thousands of patients and 1 million procedures. showing a low complication incidence.
I also agree that there are multiple routes of administration and that head to head studies have yet to be done. I have included the need for more randomized studies in the paper as you suggested.
I completely agree with comment 4 and I revised the manuscript. This suggestion was helpful in improving the manuscript.
Comment number 5 was excellent and the manuscript has been edited to reflect your point.
Reference 12 access link has been revised thank you for bringing that to my attention.
I have replaced reference14 with the following peer reviewed publications:
1. Gouras G et al. Testosterone reduces neuronal secretion of Alzheimer’s b-amyloid peptides. PNAS 2000; 97:1203
2. Bianchi V. The Anti-Inflammatory Effects of Testosterone. J Endocr Soc. 2019 Jan 1; 3(1): 91–107.
3. Zitzmann M, Weckesser M, Schober O, Nieschlag E. Changes in cerebral glucose metabolism and visuospatial capability in hypogonadal males under testosterone substitution therapy. Exp Clin Endocrinol Diabetes. 2001;109(5):302-4.
As regards reference 26, I have removed it from the manuscript and revised the paper accordingly.
Thank you for your comments on references 29, 30. I have revised the paper and included the following reference using testosterone only in the treatment of breast cancer. Boni C. Therapeutic Activity of Testosterone in Metastatic Breast Cancer. Anticancer Research 2014;34: 1287-90
Again thank you for the opportunity to revise my paper.
Gary Donovitz, MD
Reviewer 3 Report
The author discussed an important and interesting topic of the use of testosterone in women in case of its deficiency. He presented his thoughts in an interesting way supported by literature.
Please make a minor correction: Line 154: it should be 2022
Author Response
Thank you for reviewing my paper and spending the time to help with corrections that make it better. I have changed the year to 2022 as you suggested.
I appreciate your help and support.
Gary Donovitz MD
Reviewer 4 Report
In general, it is well written comment about the testosterone use for the women’s health, although many controvercial issues are still present. The current comment is one of the acceptable reference.
One question is still raised about the author’s conclusion: what does physiologic doses of testosterone mean? What is the reference about the physiologic level in women’s body? Although it is a safe description, but not scientific description. Please revise it and additionally, please add information about the natural course of testosterone decline in women. Finally, please add data about the trials addressing the dosage use of the testosterone.

Author Response
Thank you for taking the time to review my manuscript. I appreciate your feedback and comments.
I agree with you that physiologic/pharmacologic treatment of testosterone deficiency was confusing. I have replaced it with personalized individualized dosing and revised the manuscript.
In fact there is no normal testosterone levels for women was well documented in this reference: Bachmann G. et al. Female androgen deficiency: the Princeton Consensus statement on definition, classification, and assessment 2002; Fertility and Sterility;77:660-665. ( It is included in the manuscript)
I have included how we dosed women in the long term complication study reference13. This suggestion was an excellent addition to the manuscript.
Thank you again,
Gary Donovitz MD
Round 2
Reviewer 1 Report
Check line 101, does not make sense.
I would suggest changing the wording “I believe” to more considered statements.
I am glad pleased that allegations of gender bias have been removed as many feel passionate about gender bias in the reverse direction.
Author Response
Thank you for taking the time to review my revised manuscript. Per your suggestion, I edited that sentence. I appreciate your feedback.
Sincerely,
Gary Donovitz, MD
Reviewer 2 Report
I consider that the revised version of the
manuscript does not properly answers my
multiple critics and reservations.
Author Response
Thank you for taking the time to review my revised manuscript. I responded to all of your comments one by one. I considered all of your suggestions and revised the manuscript. Please let me know if any specific comment needs more clarification.
Thank you agin,
Gary Donovitz, MD